

# Effect of femtosecond laser pulse repetition rate on nonlinear optical properties of organic liquids

Sandeep Kumar Maurya, Dheerendra Yadav and Debabrata Goswami

Department of Chemistry, Indian Institute of Technology, Kanpur, India

## ABSTRACT

The effect of the repetition rate of femtosecond laser pulses on the two-photon absorption and nonlinear refraction of pure organic liquids is presented using the conventional Z-scan technique. Such a study provides a way to determine the nature of light-matter interaction, explicitly enabling the identification of the linear versus nonlinear regimes. Based on the type of light-matter interaction, we have identified the thermal load dissipation time for the organic liquids. Our experimental results are in good agreement with the theoretically calculated decay time for the dissipation of thermal load.

## INTRODUCTION

Over the past few decades, the characterization of Nonlinear Optical (NLO) response of materials has been of great interest to the scientific community due to their potential applications in various fields (*LaFratta et al., 2007*; *Spangler, 1999*; *Fisher et al., 1997*; *Zipfel, Williams & Webb, 2003*; *Helmchen & Denk, 2005*; *Parthenopoulos & Rentzepis, 1989*; *Kawata & Kawata, 2000*; *Lin et al., 2003*). Pulsed laser sources allow these nonlinear optical processes to occur at low laser power. In light–matter interaction, the transformation of CW laser into pulsed laser gives fascinating space–time control over NLO response. Pulse laser properties (pulse width, wavelength, repetition rate, chirp, and polarization) have shown modulation of a NLO property of materials. Pulse width (*Yi et al., 1998*) and wavelength (*Srinivas et al., 2001*; *Szeremeta et al., 2013*) have demonstrated a great influence on the nonlinear properties like nonlinear refraction (*Major et al., 2004*), nonlinear absorption (*Golubev et al., 2011*), and nonlinear scattering (*Clay, Wostyn & Persoons, 2002*; *Matcher, Cope & Delpy, 1997*), depending on the properties of the target material. Depending on the molecular property, one can get the order of magnitude difference in the two-photon absorption cross-section (TPACS) by the varying pulse width (*Xu & Webb, 1996*).

In the recent past, the effect of polarization (*Vivas et al., 2012*; *Chen-Hui et al., 2012*; *O'Dowd et al., 2008*) and repetition rate (*Yuksek et al., 2008*; *Gorling, Leinhos & Mann, 2003*; *Yin & Agrawal, 2007*; *Liu & Li, 2013*) of laser pulses on the nonlinear optical properties of materials have been of great interest to the scientific community. Nonlinear Refraction (NLR) of pure liquids (isotropic media) has not been studied thoroughly as far

Corresponding author
Debabrata Goswami,
dgoswami@iitk.ac.in

as the repetition rate of the pulse laser is concerned (*O'Dowd et al., 2008*). However, all materials exhibit focusing and defocusing effects (*Sheik-Bahae et al., 1991*) upon interaction with a pulsed laser, which, in turn, depend on the repetition rate of the laser pulses. In the recent past, a vast study on the effect of high repetition rate (HRR) laser pulses have been reported, where thermal load due to a train of laser pulses overwhelm pure single pulse NLO effects. There have been several attempts to obtain the pure NLO property free from thermal inhibition from intense laser pulses. In recent years, optical chopper and pulse picker are being used to manage the cumulative thermal effect in the optical medium (*Gnoli, Razzari & Righini, 2005*; *Nag, De & Goswami, 2007*; *Davila Pintle, Lara & Iturbe Castillo, 2013*; *Edziah et al., 2011*; *Shehata & Mohamed, 2019*). However, these techniques have their limitations due to their operating principles. For instance, use of optical chopper in the determination of NLO property minimizes the cumulative thermal effect from intense HRR laser pulses (*Shehata et al., 2019*; *Nag, De & Goswami, 2009*) with an enhancement in the signal to noise ratio (SNR) for the nonlinear optical effect.

Mechanism of focusing and defocusing phenomenon in materials can be understood by the fact that a molecule gets excited upon interaction with light. It may then lose energy by radiative and non-radiative pathways, resulting in thermal load near the illuminated volume. Thermal decay times are in the order of milliseconds or higher. This results in the heating of the material leading to a change in its refractive index ($n_2$). Depending on the change in the refractive index in the material, there can be a positive refractive index change (convergence of light) known as 'self-focusing' or a negative refractive index change (divergence of light) known as 'self-defocusing'.

The Z-scan technique is one of the most straightforward and conventional methods to study the nonlinear optical property of materials, which was proposed by M. Sheik-Bahae and E.W. Van Stryland (*Sheik-Bahae, Said & Van Stryl, 1989*) for the determination of nonlinear refraction index. They have shown the reversal of sign in the nonlinear index of refraction with the variation of wavelength (*Sheik-Bahae et al., 1990*). The Z-scan technique can determine the two-photon absorption cross-section ($\sigma_2$) and nonlinear index of refraction ($n_2$), which are directly related to the imaginary and real part of the third order nonlinear susceptibility, respectively. In general, the determination of NLO properties is performed in a conventional and readily available solvent (like methanol, dichloromethane, dimethyl sulfoxide, etc.), which must be fully transparent to the laser frequency. However, these solvents also go to the non-resonant thermal heating with ultrashort laser pulses. The primary aim of this paper is to address the associated thermal effect with the interaction of femtosecond laser pulses to organic liquids. The choice of our organic liquids solemnly depends on the nonlinear absorption characteristic to study the influence of thermal load on nonlinear refraction and nonlinear absorption.

Herein, we report the effect of repetition rate on the nonlinear optical property of methanol ($CH_3OH$) and carbon disulfide ($CS_2$) as an organic solvent. Organic liquids were chosen based on their nonlinear optical properties around the central wavelength of the laser. While $CH_3OH$ exhibits only nonlinear refraction, $CS_2$ exhibits both Two-Photon Absorption (TPA) and NLR upon interaction with a femtosecond laser pulse at 800 nm.

This study was aimed to investigate the influence of repetition rate on NLR and to determine the thermal decay time ($\tau$) for both the organic liquids.

## MATERIALS & METHODS

Our femtosecond experimental setup involves a tunable repetition rate Ti:Sapphire Regenerative Amplifier System (Spitfire-Pro; Spectra-Physics Inc., Santa Clara, CA, USA) that is seeded with a Spectra-Physics Mai-Tai Ti:Sapphire oscillator, having 82 MHz repetition rate with wavelength tunability of 780–820 nm. The maximum repetition rate experiments were performed with the broadband Mai-Tai oscillator at 82 MHz. The amplifier cavity with two Pockel's cells makes the different repetition rates possible as an interplay of the time of injection of the seed pulse in the amplifier cavity and the dump time that extracts the amplified pulse out of the cavity. Consequently, we were able to generate amplified laser pulses of different repetition rates, such as 5, 10, 20, 25, 50, 100, 200, 250, 333, 500, and 1,000 Hz. Figure 1 shows a schematic of the experimental setup, which describes closed aperture Z-scan (CAZS) and open aperture Z-scan (OAZS) schemes. Femtosecond pulses from the amplifier were focused using a 20 cm lens onto a one mm quartz cuvette. The cuvette contains a liquid sample for Z-scan experiments, where the path length of the sample cell satisfies the condition that the cell-length is less than the Rayleigh range of the focusing lens. Rayleigh range in our setup was 1.3 mm at 800 nm, as the beam waist at the focal point was 18 μm. We scanned the sample through the focal point of the lens using a motorized translation stage (Newport Inc. model ESP 300), whose minimum step size is 0.1 μm. This allows for a smooth intensity scan of the sample. To ensure the least interference from material dispersion from femtosecond pulses, all the mirrors used in our setup were zero-dispersion mirrors under our experimental conditions.

The transmitted beam through the sample is focused with a lens of 10 cm focal length into a UV-enhanced PDA100A-EC-Si Switchable Gain Detector. The peak-to-peak values from the photodiode are measured with an LTM-354 LeCroy oscilloscope, which is triggered by laser frequency for each repetition rate. The reference signal from detector $D_1$ was also recorded for the normalization of the signal from detectors $D_2$ and $D_3$. The delay stage and the oscilloscope are interfaced with the computer using a GPIB card (National Instruments Inc.), and the data is acquired using LabVIEW programming. After acquiring the data, results were fitted in the following equation, for the open aperture z-scan data (*Nag, De & Goswami, 2009*):

$$T(z) = 1 - \frac{\beta I_0 L}{2\sqrt{2}\left(1 + \frac{z^2}{z_0^2}\right)} \tag{1}$$

where $\beta$, $I_0$, $L$ and $z$ is the two-photon absorption coefficient, the intensity of femtosecond laser pulse at the focus in $Jcm^{-2}$, sample thichness and sample position respectively. At different z positions, transmittance T is calculated, and since all the other parameters are known, $\beta$ can be calculated easily. Subsequent to obtaining the values of $\beta$, the TPA cross-section of chromophore ($\sigma_2$, in units of $GM$, where $1GM = \frac{10^{-50}cm^4.s}{photon.molecule}$) is generated from the expression: $\sigma_2 = \beta h\nu^* 10^3 / Nc$, where $\nu$ is the frequency of the incident laser beam,

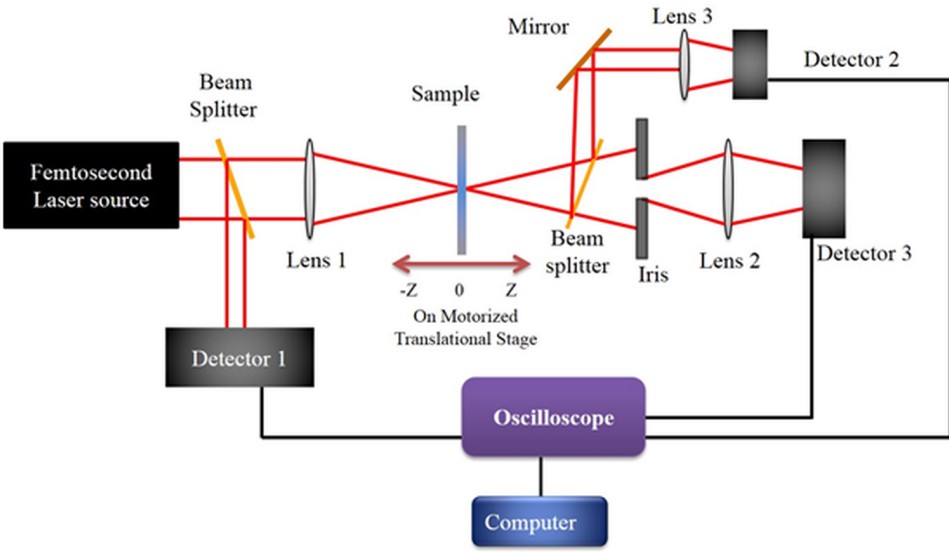

**Figure 1** **Experimental Setup.** Schematic of the experimental setup for open aperture and close aperture z-scan technique. While the amplified laser output at various repetition rate was used directly, the laser oscillator output for the experiment was additionally chirp-compensated with a prism pair.

$N$ is Avogadro constant, $c$ is the concentration of the molecule in respective solvents, while Eq. 2 was used to fit the close aperture z-scan data (*Nag, De & Goswami, 2009*; *Sheik-Bahae, Said & Van Stryl, 1989*):

$$T(z, \Delta\varphi_0) = 1 + \Delta\varphi_0 \frac{4x}{(x^2+1)(x^2+9)} \qquad (2)$$

where, $x = z/z_0$, $\Delta\varphi_0(t) = k\Delta n_0(t)L$, and $k = 2\pi/\lambda$ is known as wave vector. $\lambda$ denotes the wavelength and $\Delta n_0(t) = n_2 I_0(t)$, where, $n_2$ is the coefficient of nonlinear refraction in the unit of $m^2/W$.

## RESULTS AND DISCUSSION

The Z-scan experiment for the organic liquids, $CS_2$ and $CH_3OH$, was carried out at 800 nm, for which, both the liquids were transparent at low laser pulse power. However, at high powers of the femtosecond laser pulse, $CS_2$ exhibits TPA. Both open aperture and close aperture Z-scan techniques were used for studying the influence of repetition rate on NLR and Two-photon absorption cross-section (TPACS) of the organic solvent. To achieve the desired nonlinear response from these liquids, we have used average laser power of 145 GW/$cm^2$ where methanol exhibit only NLR and $CS_2$ exhibit both NLR and TPA. OAZS trace of $CS_2$ is shown in Fig. 2A, while the CAZS traces for both the organic liquids are shown in Fig. 2B at one kHz repetition rate. These organic liquids exhibit a positive nonlinear index of refraction (valley-peak) at one kHz, as compared to a negative nonlinear index of refraction (peak-valley) at 82 MHz repetition rate of the laser pulse, as shown in Fig. 3. The reversal of sign in the $n_2$ arises from the cumulative thermal load of the train of femtosecond laser pulses with very short pulse-to-pulse intervals (*Harzic et al., 2002*).
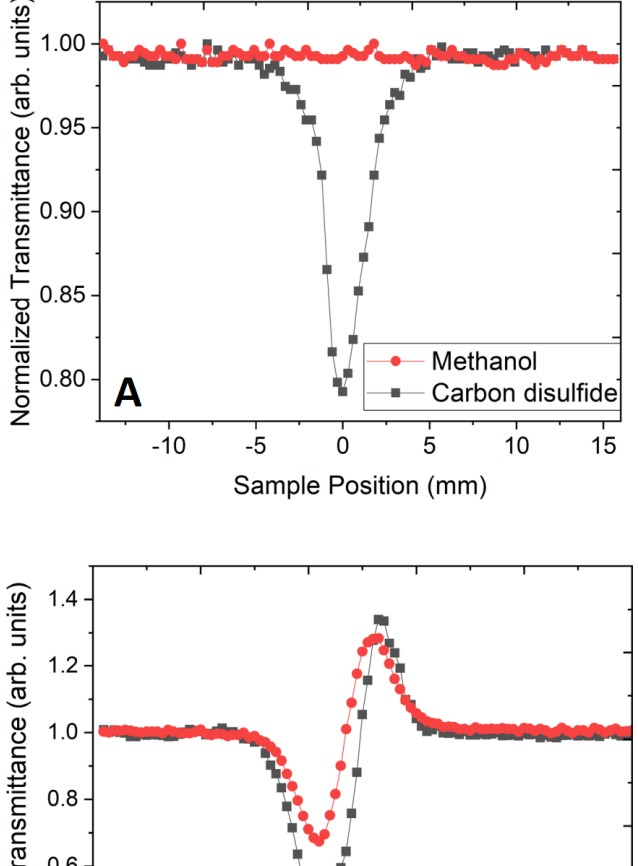

**Figure 2 Open and close aperture z-scan.** (A) Open aperture z-scan trace and (B) close aperture z-scan trace of methanol and carbon disulfide at the power of 145 GW/cm$^2$ using 45 fs pulse width laser pulse at 800 nm.

Thermal load generates a diverging lens due to thermal expansion (hence negative $n_2$) and since both methanol and carbon disulfide have negligible absorption at 800 nm, their $n_2$ values are similar.

The impact of thermal load can be further understood from the concept of acoustic transit time ($\tau_{ac}$), which is a ratio of beam waist at the focus and, the speed of sound in the medium (*Kovsh, Hagan & Van Stryland, 1999*). The beam waist of our Z-scan setup at the focus was found to be 18 μm, and the speed of sound happened to be 1,103 m/s and 1,149 m/s for methanol (*Dávila & Trusler, 2009*) and carbon disulfide (*Herzfeld & Litovitz, 1959*). Hence, the calculated $\tau_{ac}$ for the respective liquids are 16.3 ns and 15.7 ns. This indicates that pulse-to-pulse separation in MHz regime is in comparison with acoustic transit time,

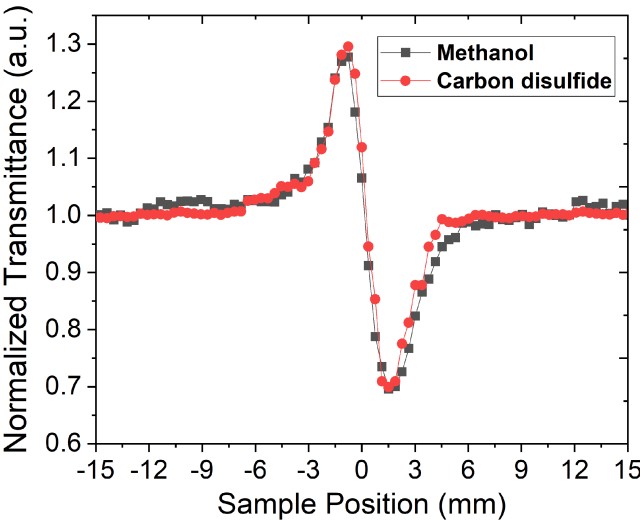

**Figure 3** **Close aperture z-scan trace at high repetition rate laser.** Close aperture z-scan trace of $CH_3OH$ and $CS_2$ at 800 nm with 40 fs chirp-compensated laser pulses from the laser oscillator with an average power of 11 GW/cm² at the focus of 20 cm lens and a repetition rate of 82 MHz.

whereas in kHz regime, the separation between two consecutive pulses is in the orders of a couple of magnitudes longer than acoustic transit time in both the organic liquids. Hence, the thermal load is expected to occur in organic liquids with MHz repetition rate laser, which in turn, results in the thermal lens effect in these liquids. This eventually leads to the negative nonlinear index of refraction in MHz, whereas the positive nonlinear index of refraction was observed in one kHz laser pulses. Apart from the acoustic transit time of these organic liquids, thermal conduction time also plays an important role in the negative index of nonlinear refraction, which is given by the following equations (*Falconieri & Salvetti, 1999*; *Gordon et al., 1965*):

$$\tau = \frac{\omega_0^2}{4K} \tag{3}$$

$$K = \frac{\kappa}{\rho c_p} \tag{4}$$

where $K$ is the thermal diffusivity, $\omega_0$ is the beam waist at the focus of the lens, $\rho$ is the density of the liquid, $c_p$ isspecific heat capacity and $\kappa$ is the thermal conductivity of the liquid. Given the above mentioned parameters for methanol : $\rho = 0.791 \text{ gcm}^{-3}$, $c_p = 2.56 \text{ Jg}^{-1}\text{K}^{-1}$, $k = 0.203 \text{Wm}^{-1}\text{K}^{-1}$, and for carbon disulfide ($CS_2$): $\rho = 1.263 \text{ gcm}^{-3}$, $c_p = 0.995 \text{ Jg}^{-1}\text{K}^{-1}$, $k = 0.16 \text{ Wm}^{-1}\text{K}^{-1}$, and $\omega_0 = 18 \text{ µm}$, the calculated thermal conduction time under our experimental condition is 0.82 ms for methanol and 0.64 ms for $CS_2$. Thus, the thermal lens is apparent in these liquids at a high repetition rate of the laser pulses, resulting in a self-defocusing effect with negative $n_2$ in these liquids. On the other hand, at one kHz, the system exhibits a self-focusing effect with positive $n_2$. The calculated value of the $n_2$ with the fitting of the CAZS trace is $1.2 \times 10^{-7}$ cm²/GW for methanol, and $2.7 \times 10^{-6}$ cm²/GW

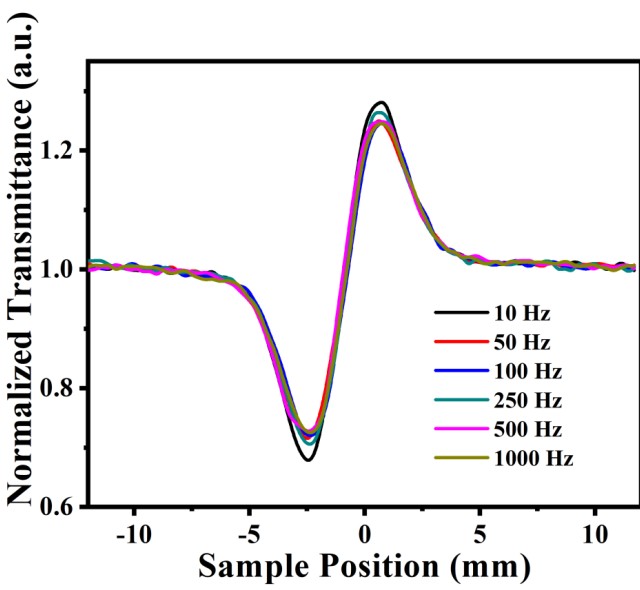

**Figure 4 Close aperture z-scan trace at different laser repetition rates.** Close aperture z-scan trace of methanol at 800 nm from an amplifier at different repetition rate having laser pulses ranging from 45 to 110 fs, where average power of the laser pulse was kept constant.

for $CS_2$ at 800 nm, for 45 fs pulse at one kHz repetition rate. TPA coefficient for $CS_2$ was found to be: $4 \times 10^{-2}$ cm/GW. These results are in good agreement with the values found in the literature (*Couris et al., 2003*; *Ganeev et al., 2004*; *Wilson et al., 2012*).

Laser power was adjusted in such a way that there was no distortion in the pulse profile of the laser while propagating through the medium. Under the same experimental condition, by keeping the constant intensity of laser pulses, CAZS experiment was carried out for both liquid samples at each repetition rate varying from one kHz to 10 Hz. CAZS trace at variable repetition rate is shown in Fig. 4 for the methanol, whereas OAZS trace for $CS_2$ is shown in Fig. 5. CAZS study with varying repetition rate show a measurable change in the peak-valley transmittance in both the liquids, which eventually resulted in the nonlinear index of refraction as presented in Figs. 6A and 6B for methanol and $CS_2$, respectively. The change in $n_2$ was observed with the variation of the repetition rate of the laser pulses. Modulation in the nonlinear index of refraction with the variation of the repetition rate of the laser is a consequence of thermal load on the liquids due to the single laser pulse interaction, which generally decay in milliseconds to second timescales depending on the thermo-optical property of the liquids. Typically, the mechanism of the thermal decay consists of conduction, convection and radiative processes. Out of these possibilities, thermal decay by convection varies from situation to situation irrespective of materials and is hard to predict. As mentioned earlier, probable contribution to the change in $n_2$ could due to the influence of thermal conductivity. Our study on the nonlinear index of refraction with a variable repetition rate provides some opportunity to implement the pulse-to-pulse separation of the laser to determine the thermal decay in liquids. As shown in Fig. 6, we have implemented the pulse-to-pulse separation ranging from 1 ms to 0.1 s,

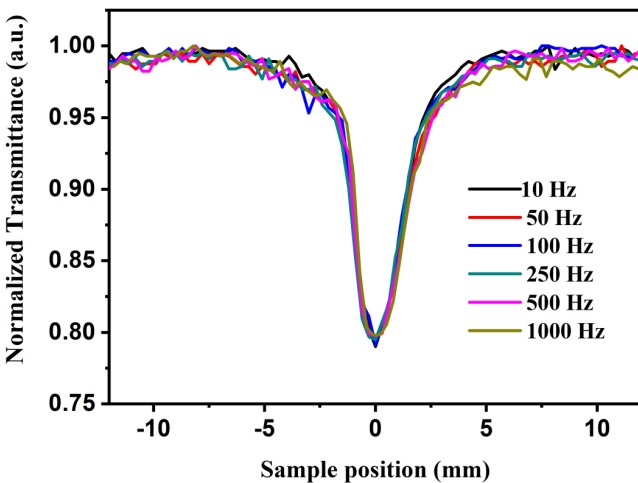

**Figure 5** **Open aperture z-scan trace at different repetition rates.** Open aperture z-scan trace for carbon disulfide at 800 nm with 45 fs pulse width and average power of 145 GW/cm$^2$ for carbon disulfide at one kHz from Ti:Sapphire regenerative amplifier. Note that methanol does not show any open aperture z-scan, as shown in Fig. 4.

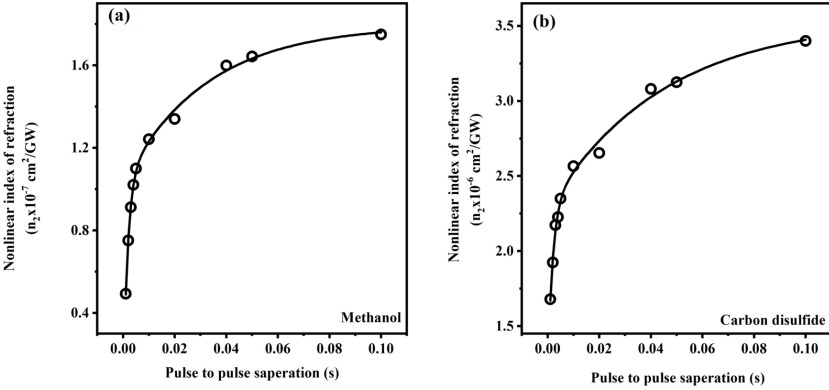

**Figure 6** **Variation of nonlinear index of refraction with pulse to pulse seperation.** Variation of nonlinear index of refraction as a function of the pulse to pulse separation for femtosecond laser pulses (ranging from 45 to 110 fs) for (A) methanol ($CH_3OH$) and (B) carbon disulfide ($CS_2$) in close aperture z-scan.

which has a direct influence on the nonlinear index of refraction. The variations obtained in the nonlinear index of refraction have been fitted with a double exponential model, which resulted in the two-time constants $T_1$ and $T_2$. The time constant $T_1$ can be accounted for thermal decay time, which represents the time required for the complete removal of thermal effect due to femtosecond laser pulses. On the other hand, time constant $T_2$ somewhat associated with the effect of broadening of pulses (as shown in Fig. 7, which was measured using noncollinear autocorrelation technique alongside the measurement of z-scan) along with the effect of convection and other non-radiative time decays. Since, heat accumulation on transparent liquid due to intense laser pulses can decay through

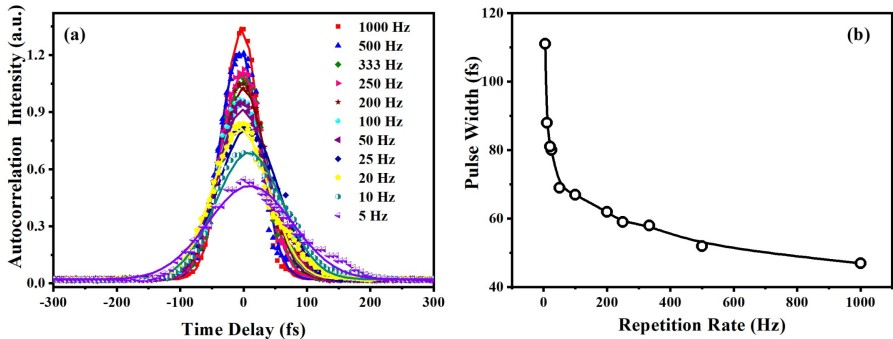

**Figure 7** **Variation of measured pulse width at different repetition rates.** (A) Pulse width measurements (45 to 110 fs) at a different repetition rate of the laser amplifier at 800 nm, and (B) variation of the pulse width with the laser repetition rate.

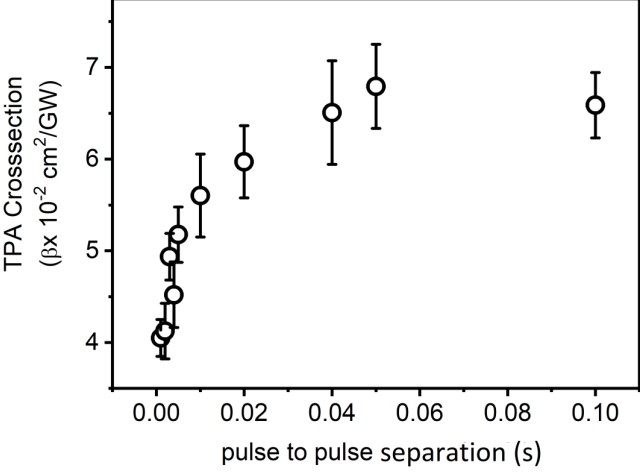

**Figure 8** **Variation of TPACS with pulse to pulse separation.** Variation of TPA cross-section as a function of the pulse to pulse separation of the laser pulses (ranging from 45 to 110 fs) at 800 nm for carbon disulfide ($CS_2$) in open aperture z-scan.

heat conduction and convection (*Kumar, Khan & Goswami, 2014*) whose time scale can be orders of magnitude different depending on the resonant (highly absorbing liquid) and nonresonant (non-absorbing liquid) thermal accumulation (*Kumar, Dinda & Goswami, 2014*). The time constant associated with $T_2$ were further analyzed and attributed to the broadening of the pulse due to the support of the TPACS variation as a function of the pulse to pulse separation (Fig. 8), which were found to be same as the time constant obtained from the variation of NLR. As shown in Fig. 8, it is not possible to discern the trend even on including all possible causes of errors that may creep in the calculation of the TPACS.It is observed from the CAZS experiments that the thermal load is apparent even at a low repetition rate of the laser pulse, such as, one kHz, which further diminishes, as the pulse-to-pulse separation increases with the decrease of repetition rate. A pictorial

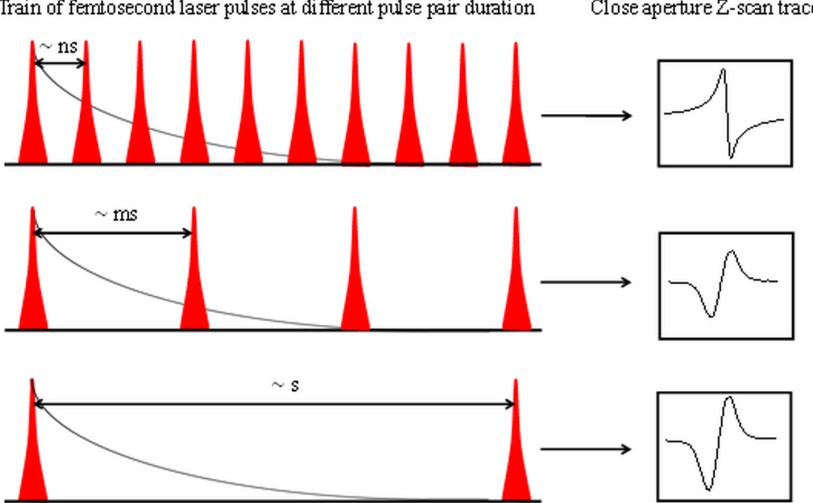

Train of femtosecond laser pulses at different pulse pair duration    Close aperture Z-scan trace

**Figure 9** **Image of close aperture z-scan traces as a function of the variation of pulse-to-pulse separation.** Schematic profile of close aperture z-scan traces as a function of the variation of pulse-to-pulse separation where x-axis represents the sample position and the y-axis represents the normalized transmittance.

**Table 1** **The thermal decay time constant as a variation of pulse repetition rate.** A double exponential fit of the change in transmittance with respect to pulse pair separation provides the calculated thermal conduction time. The thermal decay time constant as a variation of pulse repetition rate is of the same order as the calculated thermal conduction time.

| Double exponential decay fit | $Y = Y_0 + A_1 \exp\left(-\frac{t}{T_1}\right) + A_2 \exp\left(-\frac{t}{T_2}\right)$ | | Theoretically calculated time constant from Eq. 3 |
|---|---|---|---|
| | $T_1$ (ms) | $T_2$ (ms) | $T_1$ (ms) |
| Methanol | $1.74 \pm 0.3$ | $32.1 \pm 4.4$ | 0.82 |
| Carbon disulfide | $1.92 \pm 0.5$ | $44.8 \pm 1.2$ | 0.64 |

representation of this variation is shown in Fig. 9. Thus, for the first time, we report here, the thermal decay time constant probed by the conventional z scan technique as a variation of pulse repetition rate, which was found to be on the order of the calculated thermal conduction time as tabulated in Table 1.

## CONCLUSIONS

Here, we have presented the study of the effect of repetition rate on the nonlinear index of refraction of methanol ($CH_3OH$) and carbon disulfide ($CS_2$), using femtosecond laser pulses at 800 nm. This study indicates that the nonlinear index of refraction is prone to the pulse-to-pulse separation as compared to the nonlinear absorption, which does not show any significant change with respect to the repetition rate of the laser. This result also exhibits that working with laser pulses as low as 100 Hz or lower (*Singhal, Dinda & Goswami, 2017*) can give rise to almost pure nonlinear optical properties. Experimentally calculated thermal decay time from this study was found to be in good agreement with

the theoretically calculated thermal conduction time for both the organic liquids under consideration.

## ACKNOWLEDGEMENTS

All the authors acknowledge Ms. S. Goswami for extensive language editing of this article.

### Funding
This work was supported by the Intramural Individual PI Funds of SERB (DST, Govt. of India); STC Grant of ISRO (Govt. of India); and MeitY Grant of the Min. of IT (Govt. of India). The funders had no role in study design, data collection and analysis, decision to publish, or preparation of the manuscript.

### Grant Disclosures
The following grant information was disclosed by the authors:
Intramural Individual PI Funds of SERB (DST, Govt. of India).
STC Grant of ISRO (Govt. of India).
MeitY Grant of the Min. of IT (Govt. of India).

### Competing Interests
Debabrata Goswami is an Academic Editor for PeerJ.

### Author Contributions
- Sandeep Kumar Maurya and Dheerendra Yadav performed the experiments, analyzed the data, prepared figures and/or tables, performed the computation work.
- Debabrata Goswami conceived and designed the experiments, contributed reagents/materials/analysis tools, performed the computation work, authored or reviewed drafts of the paper, approved the final draft.

### Data Availability
Th raw data is available as a Supplemental File. All the experiments presented and the data collected were at our femtosecond laser laboratory at IIT Kanpur. Supplementary information along with this article also carries the necessary details from the same Femtosecond Laser Laboratory of IIT Kanpur.

### Supplemental Information
Supplemental information for this article can be found online at http://dx.doi.org/10.7717/peerj-pchem.1#supplemental-information.

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
