# Peer review of "Effect of femtosecond laser pulse repetition rate on nonlinear optical properties of organic liquids"

_PeerJ Physical Chemistry, doi:10.7717/peerj-pchem.1_

## Round 0.1 · original submission · Major Revisions

Thank you for submitting your paper to 'PeerJ Physical Chemistry'. I expect your paper is worth publishing in PeerJ. But I also believe that you want to take the reviewers' comments into consideration for revision. I am looking forward to receiving your revised manuscript.

·

Basic reporting

All comments are listed in the Referee report (attached)

Experimental design

All comments are listed in the Referee report (attached)

Validity of the findings

All comments are listed in the Referee report (attached)

Additional comments

Overall the manuscript appears to be clearly and carefully written. The results presented are interesting for the field of pump probe technique. I think that the manuscript might deserve publication in Peer J Physical Chemistry after some points are dealt with and some missing details are added prior to publication.

Reviewer 2 ·

Basic reporting

There are some typo that need to be corrected. For example:
In line 57: "nonlinear optical " should be abbreviated as "NLO";
in line 191, "The change in *2 was observed" should be "The change in n2 was observed"

Experimental design

no comment.

Validity of the findings

1. The authors claim that the negative nonlinear refraction index (Fig. 3) is caused by the thermal lens effect induced by the highly repeated laser pulses. The authors should explain why the nonlinear refractive index is negative?

2. Also in Fig. 3, reversal of sign in the nonlinear index of refraction is observed in CH3OH when excited by high repetition rate laser. What's the situation with the material of CS2? More experimental results are needed.

3. As shown in Fig. 6 and Fig. 8, the authors have demonstrated the influence of the pulse-to-pulse separation on the nonlinear index of refraction. It’s reasonable that thermal effect plays an important role in the measurements. However, the repetition rate could also influence the pulse width and the averaged intensity of the pulse (as shown in Fig. 7). This influence should be considered in the measurement of the nonlinear optical response.

Additional comments

Generally, the findings in the manuscript is interesting and article is well organized. As mentioned above, some more experimental results are needed to support the author's point of view, and deeper discussion is needed to make the manuscript more readable. The reviewer suggests that the manuscript should be major revised before publication.

---

## Round 0.2 · accepted · Accept

I have found that the authors have revised the manuscript appropriately. I think that it is now suitable for publication in PeerJ Physical Chemistry. Thank you for considering to publish your manuscript in PeerJ.

·

Basic reporting

Dear Editor,

Thank you very much for the opportunity to review the revised article entitled Effect of femtosecond laser pulse repetition rate on nonlinear optical properties of organic liquids" by Sandeep K Maurya, Dheerendra Yadav, Debabrata Goswami.

The authors have made satisfactory amendments to the manuscript in response to my previous comments and concerns. Overall the manuscript reads well and has clarified the work of the authors. In my opinion the manuscript contains now all information and is suitable for publication in journal PeerJ Physical Chemistry as a regular article.

Best regards

Tarek Mohamed

Experimental design

no comment

Validity of the findings

no comment

Additional comments

no comment